# The Risk of Household Socioeconomic Deprivation Related to Older Long-Term Care Needs: A Qualitative Exploratory Study in Italy and Spain

**Georgia Casanova [1,\*], Mirian Fernández-Salido [2] and Carolina Moreno-Castro [2]**

1 Centre for Socio-Economic Research on Ageing, IRCCS-INRCA National Institute of Health & Science on Ageing, 60124 Ancona, Italy
2 Instituto de Investigación en Políticas de Bienestar Social (POLIBIENESTAR)—Research Institute on Social Welfare Policy, Universidad Valèncià, 46022 Valencia, Spain; mirian.fernandez-salido@uv.es (M.F.-S.); carolina.moreno@uv.es (C.M.-C.)
\* Correspondence: g.casanova@inrca.it

**Abstract:** Background: Older individuals with long-term care (LTC) needs represent a risk factor for poverty and socioeconomic deprivation (SED) for households. This challenge threatens the overall sustainability of health and social care systems. Spain and Italy have a robust family-based care regime with a high level of informal long-term care. Aims: This qualitative study aims to provide empirical evidence of the socioeconomic risks for Spanish and Italian households related to long-term care needs by identifying the phenomenon's main characteristics and suggestions for innovative policies and solutions. In particular, this qualitative study examines the opinions of experts and stakeholders from both countries to: (a) explore the relationship between LTC needs and household SED risk in Spain and Italy; (b) identify key associations between words and concepts, highlighting their specific characteristics in both countries; and (c) perform an in-depth analysis of the interviewees' views on designing innovative policies to support households, aimed at coping with the SED risk arising from the challenges posed by meeting the LTC needs of older people and their relatives Methods: National experts and stakeholders were involved in interviews and focus groups in both countries. A linguistic and content analysis of the transcriptions produced was performed and supported by T-LAB. Results: The SED risk for households was confirmed in both countries. Informal carers, often women, contend with a higher risk of material and social deprivation. To improve the coordination and integration strategy of public LTC, policies should contrast the SED risk of households. Moreover, an innovative integrated welfare model should consider valorizing different existing resources (e.g., informal care, neighbors, and NGOs). Conclusions: Long-term care policies aimed at promoting ageing need to pay more attention to such a risk as a structural component to be addressed and tackled with more specific and effective strategies.

**Keywords:** LTC needs; socioeconomic deprivation; innovative policies; Italy; Spain

## 1. Introduction

Population ageing and public health expenditure, especially for dependent older adults, are significant challenges for all EU Member States [1]. By 2025, there will be 825 million people aged over 65 worldwide and, by 2050, 2 billion [2]. Notably, the number of people aged 80 and over will increase nearly four-fold to 395 million [3]. The increase in public healthcare expenditure has outpaced GDP growth by 4 percent, even in high-income countries [4]. Growing needs and economic and global health crises significantly impact the sustainability of healthcare and long-term care (LTC) systems [5–7]. The literature identifies four care regimes existing in Europe [8,9]: (a) the Northern countries

(e.g., Finland, Sweden, Denmark and the Netherlands), which are based on the "Universal-Nordic regime" characterized by high and generous formal care and social support to meet a medium level of care demand and low informal care provision; (b) the central European countries (e.g., Germany, Austria, and France), based on a standard care mix where a medium/high demand for care is covered by a medium level of both informal and formal care provision; (c) the Mediterranean countries (e.g., Italy, Spain, Greece, and Portugal), which represent a traditional care regime based on family involvement identified by a high demand for care, low formal care provision, and high informal care; (d) the Eastern European countries (e.g., Hungary, Poland, and Romania), defined as an "in-transition regime" characterized by high informal care and medium formal care provision specifically aimed to cover a low level of care demand. Schulmann and Leichsenring [10] underlined how, regardless of the classification assigned, many European countries apply an informal care-oriented strategy to care but are differently supported by public contributions.

Since the 1990s, the cost containment of long-term care policies and the spread of cash benefits schemes imply a growing informal care rate and the parallel expansion of for-profit home care across Europe [11–13]. An increasing number of studies, focusing on the impact of LTC expenditure on the income of informal caregivers of dependent older people [14], set out the additional costs that individuals must defray with the onset of dependency and the difficulty in doing so given the current level of state pensions [15] or looking into the financial sustainability of an ageing population and higher levels of public healthcare expenditure [16]. A recent study has shown that, in many countries, the policies aimed at supporting informal care do not counter its negative socioeconomic impact on unpaid caregivers [17]. On the other hand, Angrisani et al. [18] have highlighted the positive impact that the availability and affordability of public services have on the cost of LTC for households. The widening inequalities in healthcare and social welfare provision have become a relevant issue in the design of sustainable development strategies [19]. Particularly in those countries with a strong family care tradition, households directly and/or indirectly provide a high level of informal LTC while also resorting to private home care workers, including migrants [20].

Italy and Spain are two of the most representative countries with such a tradition and display many similarities in terms of LTC needs, governance models, and social welfare systems [21]. In 2020, 13.2% of older people in Spain and 16.3% of older people in Italy were dependent [22] and, in view of the demographic trends in both countries, LTC will become an even more pressing issue with the passing of time, with Spain and Italy having the oldest populations in the world by 2050 [23]. In both countries, even though LTC and social welfare policymaking is the remit of their national ministries, the policies are designed, implemented, and monitored by the local authorities. In 2006, the enactment of the 'Dependency Law' (39/2006 of 14 December) marked the beginning of the reform of Spanish LTC aimed at enhancing the potential of this multilevel governance, thanks to several consultation processes involving numerous stakeholders. In Italy, a framework law reform of LTC was approved in 2023 (Law No. 33/2023) following the proposals arising from the work of a national stakeholders' network called "Patto per la non autosufficienza". In both countries, moreover, at least one in every four households is at risk of poverty [24], a plight that has also been exacerbated by the impact that the government measures adopted during the COVID-19 pandemic have had on the global economy, leading to a deep recession. Indeed, the sudden interruptions in production, the closure of borders, the plunge in international trade, the reduction in working hours, and increased layoffs have directly affected household incomes [25–27]. In this context, the "caregiver burden", defined by Simon et al. [28] (2019) as the physical, psychological, social, and economic–financial stress suffered by caregivers, has increased due to the rise in poverty and inequality resulting from the pandemic [29].

## 1.1. Related Literature

The literature underlines how the reduced power to cover the cost of care needs by older people living in deprived conditions negatively impacts their health and psychosocial well-being [30–34]. Poor older people living in rural and deprived areas meet a higher risk of mortality [35] or achieve higher levels of daily activities living (ADL) limitations [36,37]. However, single socioeconomic factors (e.g., education, gender, and marital status) have a positive impact, reducing the risk of deprived health status. Lima-Costa and colleagues [38] confirmed these results, identifying how the provision of home care increases in highly educated and high-income households. On the contrary, informal care provision results in less socioeconomically stratified effects and those that are more determined by some specific conditions of the life of older adults (e.g., living alone or not) [38]. The existing literature underlines how LTC needs to impact the socioeconomic conditions of older people and their caregivers. Being a care recipient, particularly for women, increases the individual risk of impoverishment [39–41]. The health conditions of working carers appear to deteriorate more rapidly than workers who do not provide informal care, pushing them to retire [42,43] and reducing their assets [44,45]. Moreover, other studies stress the socio-exclusion and socio-inequity risks for informal carers, often regarding women, due to the consequent limitations in working and their social life [46]. Flores et al. [47] underline the circular causal process between SE deprivation, deteriorating health, and the individual autonomy conditions of older people and their families.

A recent literature review [48] identifies several gaps in the literature by studying the relationship between LTC needs and socioeconomic deprivation (SED) household risk.

First, the literature often studies the relationship through a specific and partial aspect (e.g., out-of-pocket costs' impact on households' income or the social exclusion risk for informal carers) instead of considering the issue as a multidimensional and circular phenomenon.

Second, the informal carers and households of dependent older people are still considered secondary support measures and service recipients; care recipients remain the main target of studies. Third, the social aspects of SED (e.g., social exclusion, isolation, and poor social life) are still understudied and often not included in studies focusing on deprivation; material deprivation, poverty, and the effect on household income are the dimensions more used to analyzing the SED concept. Fourth, comparative and longitudinal studies are seldom represented by the literature on this issue. Lastly, the review study underlines the lack of qualitative studies focused on future scenarios and eventual solutions.

## 1.2. Aims and Relevance of the Study

Studying the context in which care is provided is essential when considering the perils of SED risk for households and designing innovative policies for families caring for older relatives at home in Europe [28,29].

In light of recent research having confirmed the existence of a statistical correlation between LTC needs and poverty in different European countries [30], this study aims to explore the SED risks faced by families caring for their dependent older relatives in Italy and Spain as representative countries with a family care regime, identifying the main characteristics of the phenomenon and detecting suggestions for innovative policies. In particular, this qualitative study examines the opinions of experts and stakeholders from both countries to: (a) explore the relationship between LTC needs and household SED risk in Spain and Italy; (b) identify key associations between words and concepts, highlighting their specific characteristics in both countries; (c) perform an in-depth analysis of the interviewees' views on designing innovative policies to support households aimed at coping with the SED risks arising from the challenge posed by meeting the LTC needs of older people and their relatives. In the native language of both countries, the term

"family" refers to people to whom one is related and even people who live in the same house. For this reason, the terms "household" and "family" are used interchangeably in this study. This study contributes to the literature by discussing these issues with experts and stakeholders in two countries (Spain and Italy) representative of informal care-oriented strategy in many European countries.

Moreover, this study covers the main gaps in the literature identified by the above-mentioned studies because: (a) the SED and LTC needs concepts are investigated by observing their complexity and multidimensional identity; (b) households and informal care are the main targets of the study; and (c) qualitative methods are preferred to offer suggestions on possible innovative solutions and future scenarios.

Since 2013, the European Commission has suggested employing a social innovation concept to identify new approaches. This approach strongly advocates updating LTC and welfare policies nationally and internationally [31,32]. This paper is part of the qualitative research conducted in the framework of the project "Socio-economic deprivation related to the effect of the presence of dependent older people: strategies for Innovative Policies in Europe" (SEreDIPE).

## 2. Materials and Methods

According to the aims of the study, we adopted a qualitative exploratory design based on interviews and focus groups with experts and stakeholders in Italy and Spain.

### 2.1. Participants

This study implemented a mixed strategy to select 30 participants, including experts and stakeholders, to collect their views on the issue. The experts were chosen for their academic or professional profiles, reflected by their participation in the national or international debate on LTC and SED issues. The stakeholder group included policymakers, LTC and social workers, informal carers, specialized non-governmental organizations (hereafter NGOs), and providers with experience in advocacy on LTC or SED issues. Moreover, the diverse profiles of the participants allowed us to collect views from micro, meso, and macro perspectives. Table 1 shows the distribution of participants by profile in both countries.

**Table 1.** Spanish and Italian participant profiles.

| Profiles | Spanish | | Italian | |
| --- | --- | --- | --- | --- |
| | Interviews | Focus Groups | Interviews | Focus Groups |
| Academic | 5 | 5 | 3 | 2 |
| Specialized research centers | 1 | | 3 | |
| Policymakers, professionals, and informal carers | 1 | 1 | | 1 |
| NGOs | | | 2 | 6 |
| Total | 7 | 6 | 8 | 9 |
| Total/country | | 13 | | 17 |
| Total | | | | 30 |

Following the ethics procedure approved by the Data Protection Officer (DPO) of the University of Valencia, the participants were informed about the study and data treatment by an informative sheet. Each of them signed an informative consent form on data treatment. The DPO required no other ethical requirements because the participants involved were experts, and no personal data were collected. The ethical committee approved this strategy.

### 2.2. Data Collection

From February to March 2022, 15 interviews and 2 focus groups were conducted by the first author—who is able to speak Italian and Spanish—in the national language of the participants to facilitate the discussions. All data were collected online to ensure participation while complying with the safety measures adopted during the COVID-19 pandemic. To this end, using the Zoom platform allowed for direct audio and video recording of the interviews and focus group discussions, resulting in high-quality audio transcripts. The interviews lasted 45 min on average, whereas the focus group discussions lasted approximately two hours.

The interviewer employed a predefined set of questions to conduct the focus groups and interview discussion, during which the experts and stakeholders were also encouraged to comment on the current situation in their country of origin (Italy or Spain) and to suggest possible solutions. The discussion was stimulated by the showing of findings from the quantitative part of the research [33], which was useful for debating the Spanish and Italian situation in a European framework. The set of questions and items used is detailed in Table 2.

**Table 2.** General topics and set of questions for interviews and focus groups.

| General Topic | Questions/Items |
|---|---|
| Their own opinions and/or experiences of the risks that SED posed for households providing informal care or supporting the LTC needs of dependent relatives. | - In what situation do Italian/Spanish families care for dependent older relatives? Please give us your opinion. <br> - Are these families in Italy/Spain exposed to an increased SED risk? <br> - Please give us some examples or suggestions. |
| The main characteristics of the risks that SED posed in their own country, should this be the case, including the general profile of those household members most at risk. | - What are the risks of socioeconomic disadvantage for families caring for dependent older relatives? Please describe the risks in detail. <br> - Can you draw the profile of people meeting the SED risk due to LTC needs? <br> - Where would you see a need for action to counteract the risk of socioeconomic disadvantages for affected families? |
| Innovative policy aimed at reducing the risk of SED for households attempting to cover LTC needs, identifying specific characteristics. | - If you could design one or more policies to support families/to contrast this effect, what would it/they be? Please describe it/them in detail. <br> - What would you think of as an innovation? What would the main features of this innovation be? |

### 2.3. Data Analysis

This study was based on the analyses of qualitative data collected in Italian or Spanish, depending on the country of data collection. T-LAB Plus was used to perform the analyses, for it allows for the elaboration of semantic maps representing dynamic discursive aspects and for the measurement, exploration, and mapping of co-occurrences between the main terms [49], as well as permitting users to perform various types of discourse analysis. Once the interviews had been transcribed, homogenized, and anonymized, they were analyzed to compare the results obtained in the different interviews and focus group discussions held in Italy and Spain. The analyses were conducted in Italian and Spanish by the native-speaker researchers. Moreover, T-LAB was used to perform the analyses in different languages, including Italian and Spanish. The analyses were performed in two steps. First, a preliminary linguistic analysis was carried out to support the

content analysis, word associations, thematic clusters, and word clouds for comparing the outputs from both countries, offering valuable suggestions on cultural differences and supporting the post-categorisation of variables included in the content analysis [50]. Second, a content analysis of the transcripts was performed on the T-LAB verbatims. Based on the three macro items related to the study's specific aims (detailed in Table 2), the prescriptions and verbatim transcripts were post-categorized to collect specific parts of the interviews in toll tables [51,52]. Two authors (GC and MS) carried out a cross-content analysis to reduce the effects of individual interpretation of the data, identified by the literature as the main limitation of qualitative content analysis [53]. Following their translation into English, the selected texts were summarized in tables (see Tables 3 and 4). As T-LAB requires the full anonymization of texts, the participants guaranteed the inclusion of different points of view in the content analysis. Moreover, the interviewees were numbered consecutively in each country.

**Table 3.** An overview of the risk of socioeconomic deprivation faced by caregivers and caring households.

| |
|---|
| 1.  In both countries, caregivers and caring households face the risk of SED resulting from meeting the LTC needs of dependent relatives |
| 'If I leave my job, then I'm already in a situation of poverty. What would the state give me? Well, in Spain I'd get €300/400 right now for family expenses. Or take him to an old people's home, whose cost isn't necessarily covered by the state either; so the risk of poverty is clear' (I6 SP). |
| 'It's evident that the problem exists' (I5 SP). |
| '[…] especially when discussing medium-high care needs and medium-low incomes in Mediterranean countries' (I1 IT). |
| 'Of course, lower-middle-class families definitely face this risk' (I4 IT). |
| '[…] there're two situations: those [families] that face that risk and those that don't; what intrigues me is the dividing line [between the two]. As usual, those bordering the line are in the most critical situation […] when one of the members of a family, let's say with no financial problems, becomes dependent, that's when it's plunged into crisis' (FG IT). |
| 2.  Direct and indirect care expenditure has an impact on the risk of SED |
| '[…] many of these patients' needs are often not covered by the system … these patients need many medicines that have to be paid for […]. There're also indirect costs like, for example, those relating to electricity or new nutritional needs' (I1 SP). |
| 'Households are impoverished by the cost of paying for healthcare and benefits of various kinds' (I8 IT). |
| 'On the one hand, there's the reduction in income; on the other, there's an increase in expenditure, and those who don't have savings to fall back on run the risk of being driven into the ground; so the risk exists' (I6 IT). |
| 3.  The risk of SED resulting from meeting the LTC needs of dependent relatives varies over time |
| 'Since the 1970s […], families have shrunk somewhat with progressively fewer members to share the burden of caring for dependent people' (I4 IT). |
| 'Due to the COVID-19 pandemic, there's been a reduction in income, which has led to the restructuring of these families. It has also resulted in the phenomenon of having to care for nuclear family members who're perhaps experiencing situations of dependency or who need to be cared for in some way' (I2 IT). |

4. In both countries, the phenomenon is characterized by fragmentation and marked differences between the northern and southern regions and between urban and rural areas

'In Spain, it's certainly impossible to compare the situation of older people living in villages to that of those in the same financial situation living in cities, while there's also a big difference in the availability of healthcare and social welfare services. Many very small; isolated villages have a central hospital at a fair distance. Therefore, there's the disadvantage that the elderly living in those villages may receive more care from their relatives or may even resort to the private sector, even to non-professionals' (I5 SP).

'There's a huge difference between the north and the south; essential services are provided in different contexts, involving already deprived families and situations […]. In some specific regions of Italy, some families paradoxically survive thanks to the income of older people living with them' (I4 IT).

'[…] because no one had warned me about that with this illness, for example, that it was normal for him to get up. So also these very marked regional differences, also in access to information, for example, the presence of associations. In some inland areas, you don't have access, let's say, to information, to the relational capital that can also inform you about things. These things occurred to me, as did the importance of where you lived' (FG IT).

Source: Own elaboration based on the interview and focus group transcripts

**Table 4.** Main characteristics of SED resulting from meeting the LTC needs of dependent relatives in Spain and Italy.

1. Material and social deprivation are strongly correlated as they feed back into one another

'So both types of deprivation are important; in all the interviews, isolation constantly appears as a consequence of having to devote 24 h a day [to caregiving] …' (I7 SP).

'There's a need to provide care, with which you try to cope on your own, gradually leaving work to devote yourself to care, isolating yourself and experiencing social exclusion due to a low income and barely any social life to speak of' (I1 IT).

'During a short time, there's certainly emotional suffering. There may be economic costs, difficulty in reconciling care and work, difficulty in reconciling care and your own relationships, family and friends' (I7 IT).

'Deprivation has a greater social impact on spouses and a more economical one on offspring, more often than not daughters because while obviously, wives are usually also pensioners or housewives, or if they die and there's now practically no one with a paid job, that isn't true for children' (I5 IT).

2. Job loss or abandoning the labor market is a central issue affecting the deprivation of carers

'Some impediments or barriers to professional development, to access or progress in their careers, it limits their time … sometimes they even stop working to devote themselves to care … logically it reduces the household income and also affects their well-being' (I2 SP).

'The main risk arises when someone stops working in order to be a caregiver. And this is frequently a woman with a fairly unstable job, so she loses her little income when she devotes herself to caring' (I3 SP).

'Family members who care for relatives are excluded from the job market. I would even say they're excluded from more than just the job market and social life, but that's another story. They're excluded from the job market and that's the main risk. I'm

experiencing it here in Valencia, [like] my cousin in Saragossa. He had a very good job and, in the end, he had to give it up. What are we financing ourselves with? With the properties of our parents and with what they've worked for. But what about tomorrow?' (FG SP).

'[…] many of the people who miss job opportunities or even lose real jobs tend to be adult daughters …' (I3 IT).

'It results from a gradual withdrawal from the labour market of some of the family members, who are often sons and often daughters, as we know, who go on leave, take time off work or work less hours and things that end up impacting the caregiver's income' (I6 IT).

| 3. | Long-term risk of SED for female carers |
| --- | --- |

'[…] a woman who already has a not-so-stable job, so she loses the little income she has because she has to look after someone …. In the main, they are women of low socioeconomic status aged, let's say, between 45 and 55 or 60, who may not have had a job or who have never had a stable job and, therefore, have trouble making ends meet, who are also divorced, with a broken family, who also have to look after a father or a mother. So, when they have to care for them, they're already in a precarious situation' (I4 SP).

'[…] above all risk, it isn't, it isn't only at the moment of having to provide care, but also in the long term because it's precisely these women who haven't been able to make a contribution during their lives, their working lives. And then they have to stop working, meaning that they'll be paid even less when they reach retirement age ….' (I3 SP).

'We all know they tend to be women, wives, whereas grandmothers have always done this. They were carers for free, without being paid, so this job counts for little' (FG IT).

Source: Own elaboration based on the interview and focus group transcripts

## 3. Results

### *3.1. The Risk of Socioeconomic Deprivation for Carers and Households: The Participants' "Opinions"*

Some of the participants confirmed that Spanish and Italian households were at risk of SED due to the informal care that they provided dependent relatives or because of the cost of private LTC services (I5 SP, I1 IT, I4 IT). Table 3 summarizes the main items relating to the general opinions about the existence and characteristics of the risk of SED for families providing older relatives with LTC. Based on the experts' views, it can be seen that there are many similarities in this respect between Italy and Spain. According to the interviewees, cash allowances, partnership schemes, and the public provision of services did not cover all needs, with care recipients and informal carers in both countries often having to defray the additional cost of medicine, specific treatments, and private care (I8 IT). The experts concurred that the burden of these direct costs was the primary cause of financial straits and the related risk of material deprivation. Moreover, the Spanish experts singled out the impact of maintenance costs (e.g., electricity) for home automation and installing other specialist equipment in the homes of care recipients (I1 SP). These experts' opinions emphasize the impact of macro social changes or specific phenomena on Italian and Spanish households as one of the possible factors behind SED resulting from the provision of LTC. The demographic changes occurring over the past 30 years have reduced the prevalence of large families, thus minimizing the possibility of sharing the burden of care costs (I4 IT). On the other hand, during the COVID-19 pandemic, the wage reductions affecting many families forced them to live—at least temporarily—with their older relatives, which

ensured a more acceptable level of informal care provision and better income management (I2 IT). Another two similarities between the Italian and Spanish contexts were the territorial fragmentation of the phenomenon and the differences between the north and the south. The lower availability of public healthcare and social welfare services in small, isolated towns and villages contributed to the spread of informal care or, as a last option, the engagement of private care services that often did not meet minimum professional standards (I4 IT and I5 SP). Providing informal care may even positively impact the subsistence of households. The Italian experts highlighted that in southern Italy, the benefits that the older people received had become a financial mainstay for households without any other income (I4 IT).

### 3.2. The Characteristics of the Socioeconomic Risks Faced by Caregivers and Caring Households

The analysis of the interviews and focus group discussions allowed for the identification of some of the defining traits of the socioeconomic deprivation of caregivers and caring households in Spain and Italy (see Table 4). The experts focused on the social isolation suffered by caregivers and their families, insofar as their social life was much reduced, and their material deprivation owing to the lack or absence of income (I1 IT and I7 IT). By their reckoning, SED went hand in glove, feeding back into one another, for which reason both were essential factors in informal care (I7 SP). According to one of the Italian experts, however, spouses were usually affected more by social isolation than by economic deprivation because they received pensions and disability allowances. On the other hand, offspring suffered more from material deprivation due to reduced professional income (I5 IT). The progressive withdrawal from the labor market was deemed to be the first step down the path of socioeconomic deprivation for caregivers (I2 SP, I3 IT, and I6 IT), who tended to be middle-aged women with temporary jobs and without the possibility of being relocated (I2 SP, I4 SP, and FG IT). Nevertheless, even only children deciding to care for relatives were forced to leave their jobs and to live on existing family resources (FG SP). During the interviews, some of the participants drew attention to the fact that for caregivers, the experience of precariousness and deprivation did not end when they stopped providing care because their low or non-existent participation in the job market posed a long-term risk (I3 SP).

### 3.3. Innovative Policy Suggestions

Suggestions for innovative policies on welfare and LTC aimed at countering the risk of SED for caring households are summarised in Table 5. In both countries, the experts and stakeholders agreed that there was a need to broaden LTC policies so as to include not only care recipients but also their relatives, including informal and working carers, in order to guarantee their rights (I4 ES) and to improve the system's quality as far as social needs relating to home care are concerned (I2 SP and I6 IT). The experts and stakeholders also underscored that improving public care provision in the LTC sector was the best strategy for supporting informal caregivers and caring families. In particular, there were calls for greatly extending home care hours so as to offer better coverage for care needs (FG SP and I3 IT). Furthermore, they were of the mind that soft social care (e.g., cooking and washing support) needed to be better assessed because they ensured real support for informal carers and their families (FG IT). Additionally, it was suggested that innovative LTC strategies should focus on increasing the available economic resources to counter the drop-in service quality owing to the outsourcing practices that many public institutions have resorted to (I6 IT) and to ensure sufficient cash benefits for informal carers (I6 SP). Some Italian experts stressed that integrating and coordinating services and policies was still the best way to design an innovative national LTC strategy (I1 IT). In addition, in the fragmented Italian context, a specific national guide on the level of services and the implementation of policies by local authorities may contribute to improving the standardisation of care provision at a national level (I2 IT). The reforms proposed by the participants

included reviewing access criteria, which needed to be adjusted more meticulously to socioeconomic profiles (FG SP and I1 IT), in order to more effectively identify the social welfare and care needs of dependent people and their relatives (FG IT).

Closely linked to this central vision of the household as the pivotal point around which LTC and social policies should revolve, some of the participants proposed the original idea of proximity welfare, a social welfare and public healthcare community assistance model for dependent older adults that assesses the self-management of care in households (I2 SP). Proximity welfare leverages voluntary work and other informal resources to offer a novel public response to the problem, based on the "home" as the supportive and relational context on which to build the care system. This system would not be defined by existing services but by individuals assessed according to their needs and resources (I4 IT). However, in order to function properly, such a proximity care system would require support networks other than the family, including neighbours, friends, and proximity shopkeepers (I4 IT and FG IT). In this care model, community care services and the work of general practitioners and community nurses should guarantee that the fundamental healthcare and social welfare needs of dependent people and their relatives are met (I4 IT). Some stakeholders recalled the "social guardians" experience implemented in Italy, which aims to detect the soft social needs of local people, thus contributing to such an idea of welfare (FG IT). Moreover, the proximity welfare scheme would help to change cultural beliefs on the ageing issue, promoting activities and approaches aimed at prevention (FG IT). Last but not least, to encourage welfare proximity, the experts pointed to the innovative power of specialized services, like community nurses or one-stop shops, aimed at orienting and teaching informal carers and households about managing home care, supported by public, private, and other territorial resources (I3 SP and FG SP).

**Table 5.** Suggestions for innovative strategies, policies, and best practices.

| | |
|---|---|
| 1. | Broadening the scope to include care recipients, informal and professional carers, and relatives |
| | '[…] workers must have their labour rights and the chance to pay national insurance …' (I4 SP). |
| | 'It's very important that policies be aimed at that, aid in kind, that's to say, domestic services, personal care services which in some way can be combined with family supervision' (I2 SP). |
| | 'I believe that the crux of the matter is that we continue to focus on the person who needs care; instead, we need to broaden the scope to include the family context' (I6 IT). |
| 2. | Strengthening public LTC services and care support policies, improving service coordination, and reviewing access criteria |
| | 'I think this would be a good starting point […]. Also, changing the assessment tools which are precarious, to say the least' (FG SP). |
| | 'The only solution would be for my employer not to pay me my salary but for the state to pay me part of it. Something that would allow me to remain economically stable' (I6 SP). |
| | 'The Ministry of Health cannot pay caregivers; that's the responsibility of a different ministry' (I7 SP). |
| | '[It's about] investments, that is, a greater amount of public resources should be invested in the sector, [for] outsourcing to the lowest bidder in some regions has greatly undermined service quality. So, it is definitely a greater investment of resources (I6 IT). |
| | '[…] so, radically increase the nursing and social home care hours in the region' (I3 IT). |

'Home care services should be effective. Social support for dependent persons directly in their own homes, which also means preparing meals, hygiene activities, even socialising. […] Enhancing social aspects, because that's the way to resolve the problems of families, there's no other way, look, it's the only way at present. […] the idea is to gradually reduce the economic contribution, which is the same for everyone, to try to identify the real needs of the dependent person and the caregiver, then calculate that contribution and perhaps increase the services for that person' (FG IT).

'The necessary reforms, which have been known since even before the implementation of policies, still include innovation. In Italy, access to systems should also depend on the economic situation of those requesting services. This requires action on the part of the public authorities' (I1 IT).

'Identifying as far as possible guidelines and formats that can be used by all the regions and regional health authorities or by the local authorities closest to the citizenry' (I2 IT).

3. Implementing a 'proximity welfare' policy based on enhancing existing resources and creating a network between the different stakeholders in the area, NGOs, services, households, and informal and family networks (neighbors and friends)

'We need to rethink the care system and understand that a community, public or community-based solution is needed, including through people's self-organisation' (I2 SP).

'The fundamental aspect is to empower families, to give them tools so that they really feel they're in control so that they don't feel insecure so that they don't feel at a loss to know what to do so that they don't feel lost … it's primary healthcare that has to play a leading role in empowering them … for me, the key […] is community nursing. Educational strategies for health education that, from my point of view, have to be led by community nurses' (I3 SP).

'I think that what should be done is to create a one-stop shop so that you can be assessed and offered a monthly salary package, the reinforcement of SAD [the public home care service in Spain], technical support and wheelchairs. That's how I would standardise this so that people don't have to go knocking on different doors; well, it'd be a bit like that. Like, for example, offering different modular solutions: at this level, a series of subsidies are granted; at that level, another series' (FG SP).

'[…] a proximity welfare system that promotes voluntary activities through specific home support or care services which, instead of being immersed in a purely performance-based logic, have the ability to intervene in a context that is the living environment. […] an overall focus on care policies. I'm referring to a new healthcare and social welfare system that isn't based on the system as such, on public institutions, but on the individual considered not simply as a problem but also as a resource and, therefore, aimed at the residual resources of even the weakest individuals and starting from home and the enhancement of the home and the family's ability to take responsibility for their health and care. To achieve this, of course, at this first level there would have to be a network of relationships, first of all, direct neighbourhood relationships capable of supporting them, thus preventing their isolation. Then there would be a second level including services [i.e., health centres, general practitioners and community nurses], the two main providers of support that should somehow take charge of clearly identifying the basic needs of individuals and their families in the home setting' (I4 IT).

'It's for ensuring that, apart from the social services, people living on their own can also rely on friendly neighbours or even shopkeepers to cover their needs …' (FG IT).

'The importance of networks in innovative projects. Research in the field has pointed to the importance of starting before you're old, acting to develop a culture sensitive

to the idea of widespread care, in such a way as to be prepared from a very bottom-up and not a top-down perspective' (FG IT).

'An example? Social guardians. This is preventive work, namely, we monitor situations that aren't yet particularly delicate but with which the [social] services are already familiar, […] so as to anticipate a series of services and additional aspects, in which voluntary work's also necessary, in order to avoid, or at least manage, a more problematic situation. This part of the screening, which also involves families, allows us to perform a critical analysis. I think this is one of the fundamental points' (FG IT).

Source: Own elaboration based on the interview and focus group transcripts.

## 4. Discussion

The findings of our study provide empirical evidence about the existence of socioeconomic risks for households caring for dependent older people in Spain and Italy. Material and social deprivation have both been identified as relevant risks posed by informal care provision or having to shoulder the cost of home care. This also confirms the validity of Erikson's multidimensional poverty concept [54] in an LTC context, as Casanova et al. [48] have recently proposed, which considers material and social deprivation not as separate aspects but as mutually impacting aspects of impoverishment linked to caring. However, the risk of SED for families providing informal care is a dynamic phenomenon influenced by social and historical changes that determine its characteristics. Experts have highlighted how demographic trends and the relative variations in the size and composition of families affect the family management of LTC needs [55]. On the other hand, the restrictions due to the COVID-19 pandemic and the consequent contraction of the labor market have had repercussions for the social isolation of older people and the reorganization of informal care [56]. According to the experts and stakeholders participating in our study, the financial burden resulting from meeting the LTC needs of older people is the leading cause behind the impoverishment of their family carers, with informal care provision often being seen as the cheapest solution [57–59]. Nonetheless, informal carers are usually obliged to leave their jobs or, at best, reduce their working hours, resulting in the loss of individual income and social life [60,61]. Furthermore, our findings coincide with those of studies identifying the co-living middle-aged daughter as the most common informal carer profile [62–65]. However, those studies focused on the peculiarities of the risk of SED for specific informal carer profiles, emphasizing how spouses suffer from social exclusion, isolation, and loneliness more frequently because they only have their retirement income to support themselves. Similarly, offspring of both sexes can risk impoverishment due to shorter working hours. The strong family care tradition prevailing in both countries is another main issue emerging from our results. In recent years, the literature has debated the impact of mixed care strategies on care provision models [66,67]. Experts do not question the responsible attitude of Spanish and Italian families towards caring for their dependent older relatives, thus confirming that the family care culture is firmly established in these two Mediterranean countries [68]. The results reinforce the perception that the socioeconomic effects of care provision in both countries are still issues managed internally by families as a whole rather than by single members. Families also play an essential role in designing innovative welfare models based on enhancing informal care resources and proximity relationships.

Three leading solutions emerge from the results. First, to reinforce the offered LTC services and their territorial coverture in rural and urban areas. Second, to improve policies dedicated to informal care. In this regard, the informal carers should be considered recipients of dedicated social support policies and all services and measures included in the LTC system. This solution pushes for a redesign of healthcare services dedicated to LTC. Third, to implement a new "community welfare" model based on valorizing different existing resources in a specific territory. The proposed solutions underline how in family-based care regimes, countries' strategies for improving coordination and existing

formal care services form the basis of innovative and integrated strategies for LTC and social welfare policymaking in countries with a strong family care tradition [69,70]. The innovations proposed by experts and stakeholders as a whole meet the characteristics of social innovation identified by the literature because, by aiming to better answer to LTC and social needs, they encourage the promotion of: (a) new policies or the revision of existing policies; (b) openness of the target of the beneficiary, in particular to informal caregivers; (c) actions to support the quality of life (QoL) of the beneficiaries; and (d) promotion of collaboration between interested parties and services [71–73]. In fragmented and low-provision care contexts such as those existing in Spain and Italy, the empowerment of the formal care offer or the recognition of informal care assumes an innovative power. In Spain, the socioeconomic impact on caregivers is discussed as a specific aspect of the complex and fragmented issue of dependent people and related healthcare and social welfare needs. Care recipients, households, and formal LTC services, as well as the economic impact of care provision, are all relevant aspects. In Italy, in contrast, the care burden for families is a central issue that draws attention to the care market, whereas the debate on the socioeconomic impact of care provision focuses more on the deprivation of households and individuals. The main limitations of our study have to do with the participant sample. The point of view of direct users and individuals was underrepresented because of the COVID-19 pandemic restrictions, which strongly influenced the participation of non-professional carers. Moreover, the experts and stakeholders were selected based on their expertise in the issue at a macro level. This is why, in both countries, the issue was analyzed from a national rather than a regional perspective. Comparative regional and individual studies should be performed to understand better caregiving's concrete SED effects on informal carers and households, even identifying regional differences.

## 5. Conclusions

This qualitative study explored the risk of socioeconomic deprivation for Spanish and Italian households due to the long-term care needs of one or more of their dependent relatives. Analyzing experts' and stakeholders' opinions allowed us to identify the main characteristics of the SED risk for households by identifying policy suggestions. The experts confirmed the existence of this risk for the households in both countries. The impact of private care costs, including the direct cost of migrant care workers or the cost of managing health tools at home, directly influences the income power of households. The provision of informal care determines direct and indirect impacts on SED conditions for families in both countries, reducing the working income and the social life power of informal carers, often women. The solutions suggested by the participants push for an innovative national public strategy on LTC based on integrating and coordinating services and policies and valorizing the contributions from different existing resources (including informal care, territorial private, and voluntary organizations). In this regard, an effective strategy seems to be implementing a "community welfare" approach designed locally but supported by national reform or policy. Indeed, the public strategy remains the more effective measure for contrasting the SED effects due to LTC needs. The results obtained from analyzing the Spanish and Italian contexts offer essential suggestions for European countries with informal care-oriented strategies in LTC, underlining how the policy design must consider SED's impact on household policies. The relevance of the issue requires an in-depth understanding of the literature. Comparative regional and individual studies should be performed to understand better caregiving's concrete SED effects on informal carers and households, even identifying regional differences.

**Author Contributions:** Conceptualization, G.C.; methodology, G.C. and C.M.-C.; software, C.M.-C.; investigation, G.C.; data curation, G.C. and M.F.-S.; writing—original draft preparation, G.C.; writing—review and editing, G.C.; visualization, C.M.-C. and M.F.-S.; supervision, C.M.-C.; project administration, C.M.-C.; funding acquisition, G.C. All authors have read and agreed to the published version of the manuscript.

**Funding:** The study was supported by the Marie Curie European Fellowship Grant and Horizon 2020 MSCA-IF-2019 Grant Agreement No. 888102.

**Institutional Review Board Statement:** Not applicable. The Ethics committee has not requested ethical review and approval because they were not required for a qualitative study analysing experts' opinions and not collecting personal data.

**Informed Consent Statement:** Informed consent was obtained from all subjects involved in the study.

**Data Availability Statement:** The data presented in this study are available on request from the corresponding author.

**Acknowledgments:** The authors would like to thank to Sara Santini and Flavia Piccinini for their contribution to the Italian data collection.

**Conflicts of Interest:** The authors declare no conflict of interest.

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
