# Peer review of "The Risk of Household Socioeconomic Deprivation Related to Older Long-Term Care Needs: A Qualitative Exploratory Study in Italy and Spain"

_sustainability, doi:10.3390/su152015031_

Round 1
Reviewer 1 Report
1. The informal long-term care needs is a research topic of great practical significance.
2. Through interviews and surveys, this paper has collected some information and views on family long-term care, and also sorted out relevant information.
3. It is a complicated and difficult subject to properly solve a series of long-term care problems.
4. In order to improve the academic value of the paper, it is hoped that the author will further put forward suggestions or solutions that are conducive to implementation and protection from the aspects of family, community and social policy support.
Overall, the English expression is smooth and can be further optimized.
Reviewer 2 Report
The objective of this study was to examine the association between household socioeconomic deprivation and long-term care needs among older individuals in Italy and Spain. Furthermore, the study aimed to propose innovative policy recommendations to effectively address these issues. I recommend this manuscript for publication in the Sustainability Journal. However, before accepting this manuscript for publishing, I suggest that the authors consider the following comments:
1- The aim stated in the Abstract should provide a clearer and more detailed description of the study's specific objectives.
2- In the Introduction section, the authors are encouraged to explicitly clarify the gap that this research aims to address, drawing upon relevant previous studies.
3- Please expand the abbreviations "LTC" and "SED" in their first occurrences. For example, spell out "LTC" as "long-term care"
4- To enhance the comprehensiveness of the study, consider providing an overview of long-term care systems, particularly in Europe.
Please ensure that typographical errors, such as the incorrect usage of "per cent," are corrected. The correct term is "percent," written as one word without any spaces. Additionally, please review the entire document for any other potential issues, including the word "methods" in the Abstract.
Reviewer 3 Report
Thanks for opportunity to review manuscript entitled ‘‘Risk of household socioeconomic deprivation related to older long-term care needs in familistic welfare states: A qualitative exploratory study in Italy and Spain’’ for Sustainability journal. Authors of manuscripts investigated to risk of household socioeconomic deprivation related to older in Italy and Spain cultural context. The strengths of the manuscript were that it examined variables of interest in a cultural context where appropriate using a qualitative exploratory study design. As an multivariate analyst, and researcher and reviewer in this topic, I think that almost all sections need very significant improvements. It is impossible to accept this article in this form. Because my main philosophy of reviewing a manuscript as reviewer and sometimes an article editor to improve the manuscript and not punishing the authors, I provided very specific and detailed peer review of the manuscript to increase its quality and citation potential. I hope authors of the manuscript may benefit from my review. Necessary and minor revisions reported section by section with the page and line number and when possible with suggestions.
Necessary Revisions
Title
1. Author must remove Title: from manuscript.
2. Familistic welfare states is not a common term in the literature and must be revised or removed.
3. Dot must be removed from end of title.
Abstract
4. Authors used a structured abstract. Structured abstract must consists of Background/Introduction, Method, Results, Conclusion section. Authors must rearrange abstract section as above.
5. Results section in abstract is completely missing and must be added.
6. The writing of Mrthods is wrong and must be corrected.
7. Authors must provide long name of LTC in keywords in abstract
Introduction
8. In the following sentence ‘ ‘By 2025 there will be 825 million people aged over 65 27
worldwide, and by 2050 2,000 million [2].’’ I think 2,000 million must be 2,000 billion
9. Authors must provide long name of LTC in its first use.
10. Along the Introudction section and along the manuscript, authors must construct eacjh paragraph at least 3 to 8 sentences as recommended by APA. A lot of paragraph consists of single or two sentence and this is very problematic.
11. The importance of study is very weak. Authors must provide convincing rationale for study. Specifically, authors need to answer ‘ ‘Why it is important to examine risk of household socioeconomic deprivation related to older long-term care needs in familistic welfare states: among Italy and Spain using using a qualitative exploratory study design?
12. Literature review is also very weak. Authors must give information about previous studies and their weaknesses in Introduction section.
Method
13. Following information must move at the end of Introduction ‘‘a) To explore how the relationship between LTC needs and household SED’s risk were discussed by the experts and stakeholders. b) To identify key associations between words and concepts, highlighting their specific characteristics in both countries. To perform an in-depth analysis of the participants’ views on designing innovative policies to support households aimed at combating the SED's risk arising from the challenge posed by meeting the LTC needs of older people and their relatives? ‘’
14. Research design section must completely need to rewrite. Authors need to first give information about research design of their study. And then give brief information about this research design, and lastly, they must give briefly why their study belong to this research design.
15. Autohrs must provide some demographic information about participants including gender distribution of participants, age range, and mean age of participants in each country.
16. Tablo 1 must be corrected as per APA 7 rules.
17. Authors did not give any information about ethical aspects of their study. This information must be added to Data collection section.
18. I am not able to understand what authors want to mean with ‘ ‘enquire into the following’’?
Examine?
19. Instead of item by item following must write narratively ‘‘- Their own opinions and/or experiences of the risks that SED posed for households providing informal care or supporting the LTC needs of dependent relatives. The main characteristics of the risks that SED posed in their own country, should this be the case, including the general profile of those household members most at risk. At least one innovative policy aimed at reducing the risk of SED for households attempting to cover LTC needs, identifying specific characteristics.’’
20. Following must combine with above paragraph. Single sentence is not constitute a paragraph ‘‘The discussion was stimulated by the showing of findings of the quantitative part of the research (Casanova et al., 2022b), useful to debate the Spanish and Italian situation in a European framework.’’
21. Authors must provide example questions uses during data collection process.
22. In the following sentence authors must add language after Italian or Spanish ‘‘This study was based on the linguistic and content analyses of qualitative data collected in Italian or Spanish, depending on the country.
23. The citation needed for T-LAB Plus software. It is not a commonly used software in quantitative analyses.
Results
24.. No need to following information and must be removed ‘ ‘According to the main study's aims, the results coming from the content analysis are presented in three sections exploring: a) the expert's opinions on existing SED risk in both countries, b) identifying the main characteristics of the detected SED risk and c) the suggestions for innovative policies.
25. Following ‘‘3.1The risk of socioeconomic deprivation for carers and households: the participants’ opinions’’ must correct as ‘‘3.1. The risk of socioeconomic deprivation for carers and households: the participants’ opinions’’
26. Authors must divide tables to two or three parts. Tables are overly long and it is very difficult to follow them.
Discussion
27. Authors must construct Discussion section with following subtitles Discussion, Limitations, Practical implications, Conclusion and must move all related information to related sections.
28. Practical implications section is completely missing and must be added.
29. Limitations section must significantly improve.
30. Authors used a lot of sentences that need citation in Discussion without citation. Authors must carefully read and correct this.
31. the English language editing required for this article.
Moderate editing of English language required
Reviewer 4 Report
The study investigates the socioeconomic risks associated with long-term care needs in Spain and Italy, drawing on interviews and focus groups to highlight how informal care often leads to socioeconomic deprivation. The research emphasizes the need for long-term care policies that address these risks more comprehensively. While the topic is compelling, the paper would benefit from overall improvement.
-In particular, it is essential to carefully read the paper to remove typos and English language errors.
-Before using any acronyms in the text, it's imperative to define them fully to ensure clarity for the reader.
-The section dedicated to "related literature" should stand distinct from the introduction, serving as a comprehensive review of prior scholarly works relevant to the topic. This would give context and highlight the paper's position within the larger academic conversation.
-To help readers immediately grasp the main objectives of your research, the intentions outlined in section 2, specifically points a) b) and c), should be introduced earlier, ideally within the introduction.
-Lastly, the organizational structure of the paper could benefit from a clear separation between the general summary and the final takeaways. Section 4, which provides a wrap-up, should be followed by a new section, "Conclusions". This section would not only summarize the primary findings but also delve into the broader implications of the research, offering insights into potential avenues for future studies or further exploration of the subject matter.
In particular, it is essential to carefully read the paper to remove typos and English language errors, including:
Line 17: It should be "Methods" instead of "Mrthods".
Page 5: "ruralrural-urban-rural" which likely needs correction.
Page 9: The phrase "allows us a perform a critical analysis" seems grammatically incorrect; it might be better as "allows us to perform a critical analysis".
I'd recommend going through the entire paper thoroughly to ensure no further errors are present.
Reviewer 5 Report
Most academics and politicians regard inequality as a key global challenge to sustainable development. In the public eye, excessive and unjustified inequality has a number of negative consequences: it reduces opportunities for inclusive economic growth, creates barriers to poverty reduction and upward social mobility, and contributes to social tensions. Today, the debates on the causes, consequences, dynamics and possibilities of influencing inequalities to reduce them are central in interdisciplinary discussions on global and national vectors of socio-economic development in many countries. The research is of good quality and was carried out to a good standard. The findings were correctly analyzed and presented.
Recommendations for authors:
1. Make the title of the paper shorter and clearer.
2. Make the methodology of the study more explicit.
Round 2
Reviewer 3 Report
Authors revised the article as I requested. I have no further comment regarding to manuscript.
Author Response
Many thanks for your positive answer
Reviewer 4 Report
The authors have responded appropriately to the issues highlighted in the paper. With the revisions made, the quality and readability of the paper have significantly improved. In my opinion, the paper is now ready for publication.
Author Response
Many thanks for your positive answer